# Triglyceride and Glucose Index Predicts Acute Coronary Syndrome in Patients with Antineutrophil Cytoplasmic Antibody-Associated Vasculitis

**DOI:** 10.3390/diagnostics12061486

**Published:** 2022-06-17

**Authors:** Pil Gyu Park, Jung Yoon Pyo, Sung Soo Ahn, Jason Jungsik Song, Yong-Beom Park, Ji Hye Huh, Sang-Won Lee

**Affiliations:** 1Division of Rheumatology, Department of Internal Medicine, National Health Insurance Service Ilsan Hospital, Goyang 10444, Korea; pilgyupark@nhimc.or.kr; 2Division of Rheumatology, Department of Internal Medicine, Yonsei University College of Medicine, Seoul 03722, Korea; jyp@yuhs.ac (J.Y.P.); saneth@yuhs.ac (S.S.A.); jsksong@yuhs.ac (J.J.S.); yongbpark@yuhs.ac (Y.-B.P.); 3Institute for Immunology and Immunological Diseases, Yonsei University College of Medicine, Seoul 03722, Korea; 4Division of Endocrinology and Metabolism, Department of Internal Medicine, Hallym University Sacred Heart Hospital, Anyang 14068, Korea

**Keywords:** triglyceride glucose index, acute coronary syndrome, antineutrophil cytoplasmic antibody, vasculitis, metabolic syndrome

## Abstract

This study investigated whether the triglyceride (TG) glucose (TyG) index at diagnosis could predict acute coronary syndrome (ACS) in patients with antineutrophil cytoplasmic antibody (ANCA)-associated vasculitis (AAV). The medical records of 152 AAV were reviewed. Clinical and laboratory data were collected. The TyG index was calculated by TyG index = Ln (fasting TG (mg/dL) × fasting glucose (mg/dL)/2). The cut-offs of Birmingham vasculitis activity score (BVAS) and the TyG were obtained by the receiver operator characteristic (ROC) curve and the highest tertile (9.011). The mean age was 57.2 years and 32.9% were male. AAV patients with a TyG index ≥ 9.011 exhibited a lower cumulative ACS-free survival rate than those with a TyG index < 9.011. However, a TyG index ≥ 9.011 was not independently associated with ACS in the multivariable Cox analysis. Meanwhile, there might be a close relationship for predicting ACS among the TyG index, metabolic syndrome (MetS), and BVAS. AAV patients with a TyG index ≥ 9.011 exhibited a higher risk for MetS than those with a TyG index < 9.011 (relative risk 2.833). AAV patients with BVAS ≥ 11.5 also exhibited a higher risk for ACS than those with BVAS < 11.5 (relative risk 10.225). Both AAV patients with MetS and those with BVAS ≥11.5 exhibited lower cumulative ACS-free survival rates than those without. The TyG index at AAV diagnosis could estimate the concurrent presence of MetS and predict the occurrence of ACS during follow-up along with high BVAS at diagnosis in patients with AAV.

## 1. Introduction

The 2012 revised International Chapel Hill Consensus Conference Nomenclature of Vasculitides defined anti-neutrophil cytoplasmic antibody (ANCA)-associated vasculitis (AAV) as necrotising vasculitis with few or no immune deposits, as well as small-vessel vasculitis affecting mainly capillaries, venules, arterioles, and occasionally small arteries [1]. In principle, AAV is composed of three subtypes, microscopic polyangiitis (MPA), granulomatosis with polyangiitis (GPA), and eosinophilic GPA (EGPA), according to the histopathological features including the presence of granulomas and the types of infiltrated immune cells [1,2,3]. Additionally, clinical features such as allergic components, surrogate markers suggesting GPA, ANCA positivity and type, and renal vasculitis contribute to AAV diagnosis [4].

Cardiovascular involvement of AAV includes loss of pulses, valvular heart disease, pericarditis, ischaemic cardiac pain, cardiomyopathy, and congestive cardiac failure [5]. Its frequency is relatively low compared to those of other organ involvements: up to 15%, 20%, and 50% in MPA, GPA, and EGPA, respectively [6]. Our recent study found that the overall frequency of cardiovascular manifestations at AAV diagnosis was 21.6%, whereas those of renal and pulmonary manifestations were 68.8% and 64.8%, respectively. Furthermore, 6.8% of patients experienced acute coronary syndrome (ACS) during follow-up [7]. Other recent data indicate that cardiovascular disease was the sixth leading cause of mortality due to vasculitis in the United States [8]. As such, since the mortality due to acute coronary syndrome can be one of the leading aetiologies of all-cause mortality in AAV patients, its serious clinical implication cannot be ignored. Therefore, if a novel index to predict the occurrence of ACS during follow-up is developed at the time of AAV diagnosis, it will help physicians to make decisions and enable more aggressive treatment of AAV.

The triglyceride-glucose (TyG) index, which is derived from fasting triglyceride (TG) and fasting blood glucose levels, has been proposed as a reliable surrogate marker of insulin resistance (IR). Several studies have shown that the TyG index is closely associated with traditional cardiovascular risk factors, including hyperglycaemia, hypertension, and metabolic syndrome (MetS) in a general population [9,10,11]. Recently, the TyG index was demonstrated to be useful in assessing the risk of cardiovascular disease (CVD), particularly ACS [12,13]. Therefore, it could be reasonably speculated that the TyG index could be a novel index to predict the occurrence of ACS in AAV patients; however, to date, there has been no study on the clinical implication of the TyG index in AAV patients. Hence, in this study, we investigated whether the TyG index at AAV diagnosis could predict the occurrence of ACS during follow-up in AAV patients.

## 2. Materials and Methods

### 2.1. Patients Included

One-hundred-and-fifty-two AAV patients were included in this study according to the inclusion and exclusion criteria, and their medical records were retrospectively reviewed. The inclusion criteria were: (i) patients first diagnosed with MPA, GPA, or EGPA at the Division of Rheumatology, Department of Internal Medicine, Yonsei University College of Medicine, Severance Hospital between March 2001 and May 2021; (ii) patients who fulfilled both the 2012 revised International Chapel Hill Consensus Conference Nomenclature of Vasculitides and the 2007 European Medicine Agency algorithm for the classification of AAV [1,4]; (iii) patients who had sufficiently well-written medical records to collect clinical and laboratory data, classify AAV and its subtypes, and calculate the Birmingham vasculitis score (BVAS) and five-factor score (FFS) at AAV diagnosis [5,14,15]; (iv) patients who had the results of ANCA tests, the indirect immunofluorescence assay, and the immunoassays at AAV diagnosis; and (v) patients who had the essential parameters of an equation of the TyG index and the components of MetS at AAV diagnosis. The exclusion criteria were: (i) patients with concomitant serious medical conditions such as malignancies, systemic vasculitis other than AAV, and infectious diseases requiring hospitalisation; (ii) patients who had been followed up for three months or less after the time of AAV diagnosis; and (iii) patients exposed to immunosuppressive drugs for the treatment of suspected AAV before the time of AAV diagnosis. Concomitant serious medical conditions were identified using the International Classification of Diseases (10th revision), and medications administered before and after the AAV diagnosis were reviewed by the Korean Drug Utilization Review system. The present study was approved by the Institutional Review Board (IRB) of Severance Hospital (Seoul, Korea, IRB No. 4-2020-1071), and conducted according to the Declaration of Helsinki. Given the retrospective design of the study and the use of anonymized patient data, the requirement for written informed consent was waived.

### 2.2. Variables at AAV Diagnosis

Age, sex, and body mass index (BMI) were noted as basic clinical data. Information on AAV subtypes, ANCA positivity, BVAS, and FFS were obtained as AAV-specific data. The presence of type 2 diabetes mellitus (T2DM) and hypertension was ascertained on the basis of medical history and medications. Laboratory results, in particular, those of acute phase reactants, erythrocyte sedimentation rate (ESR), and C-reactive protein (CRP), were also collected. Blood samples for serum glucose and total cholesterol, TG, high-density lipoprotein (HDL) cholesterol, and low-density lipoprotein cholesterol measurements were obtained after overnight fasting.

### 2.3. Variables during Follow-Up

All-cause mortality, relapse, end-stage kidney disease (ESKD), cerebrovascular accidents (CVA), and ACS were defined as poor prognoses of AAV. Only when they occurred at AAV diagnosis or after it were they accepted as poor outcomes of AAV in the present study. The follow-up period for all-cause mortality was defined as the time interval from AAV diagnosis to death for deceased patients and from AAV diagnosis to the last visit for surviving patients. Similarly, the follow-up period for each poor prognosis other than all-cause mortality was defined as the time interval from AAV diagnosis to the first occurrence of the poor prognosis. For patients without poor prognosis, the follow-up was defined as the interval from AAV diagnosis to the last visit. ESKD was defined as a medical condition requiring the initiation of dialysis or kidney transplantation [16]. Medications administered during follow-up were evaluated (Table 1).

### 2.4. Definition of ACS

ACS was defined as pathological conditions leading to myocardial ischaemic injuries, such as ST-segment elevation myocardial infarction (MI), non-ST-segment elevation MI, and unstable angina [17].

### 2.5. Equation of the TyG Index

The TyG index is calculated using the following equation: TyG index = Ln (fasting TG (mg/dL) × fasting glucose (mg/dL)/2) [18].

### 2.6. Classification as MetS

The diagnosis of MetS can be made only if three or more of the five components of MetS are fulfilled in the Asian population. The five components are (i) increased waist circumference (≥90 cm in men, ≥80 cm in women); (ii) high blood pressure (systolic blood pressure ≥ 130 mmHg or diastolic blood pressure ≥ 85 mmHg, or antihypertensive medication use); (iii) hypertriglyceridaemia (TG ≥ 150 mg/dL); (iv) low level of HDL cholesterol (<40 mg/dL in men and <50 mg/dL in women); and (v) impaired fasting glucose level (fasting plasma glucose ≥ 100 mg/dL) or T2DM (fasting plasma glucose level ≥ 126 mg/dL or use of medication for high blood glucose) [19,20].

### 2.7. Optimal Cut-Off of the TyG Index

The optimal cut-off of the TyG index was obtained in two ways. First, it was extrapolated by performing the receiver operator characteristic (ROC) curve analysis, and one value with the maximum sum of sensitivity and specificity was selected. Second, it was defined as the lower limit of the highest tertile of TyG as described in a previous study when no significant cut-off was set by the ROC curve [13].

### 2.8. Statistical Analyses

All statistical analyses were performed using IBM SPSS Statistics for Windows, version 26 (IBM Corp., Armonk, NY, USA). Continuous variables were expressed as medians with interquartile ranges, whereas categorical variables were expressed as numbers (percentages). Significant differences between the two categorical variables were analysed using the chi-square and Fisher’s exact tests. The Mann–Whitney U test was used to compare significant differences between two continuous variables. The correlation coefficient (r) between the two variables was obtained using the Pearson correlation analysis. The optimal cut-off was extrapolated by performing the ROC curve analysis and one value having the maximised sum of sensitivity and specificity was selected. The relative risk (RR) of the cut-off for high AAV activity was analysed using contingency tables and the chi-square test. Comparison of the cumulative survival rates between the two groups was performed using the Kaplan–Meier survival analysis with the log-rank test. The odds ratios (ORs) at AAV diagnosis and the hazard ratios (HRs) during the considerable follow-up duration were obtained using multivariable logistic regression analysis and multivariable Cox hazards model analysis. In this study, the variables that showed a statistical significance of *p* < 0.1 in the ROC curve and the univariable analysis were included in the multivariable analysis to find more variables with clinical implications. However, in the multivariable analysis, statistical significance was defined as *p* < 0.05. *p*-values less than 0.05 were considered statistically significant.

## 3. Results

### 3.1. Characteristics

At AAV diagnosis, the mean age was 57.2 years and 50 patients (32.9%) were male. Eighty-four, 35, and 33 patients were diagnosed with MPA, GPA, and EGPA, respectively. ANCA was detected in 119 patients (78.3%). The mean BVAS, FFS, ESR, and CRP were 12.0, 1.0, 56.0 mm/h, and 7.0 mg/L. As for the variables composing the TyG index, the mean fasting glucose and TG were 103.0 mg/dL and 114.5 mg/dL, respectively. The mean TyG index was 8.8. During follow-up, 12 patients (7.9%) died, and 48 (31.6%) experienced a relapse. ESKD, CVA, and ACS occurred in 31 (20.4%), 12 (7.9%), and 12 patients (7.9%), respectively. Glucocorticoids were administered to 144 patients (94.7%), and the most commonly used immunosuppressive drug was cyclophosphamide (51.3%) (Table 1).

### 3.2. Correlation of the TyG Index with Continuous Variables

The TyG index was significantly correlated with BMI (r = 0.222, *p* = 0.006), HDL-cholesterol (r = −0.182, *p* = 0.029), and LDL-cholesterol (r = −0.236, *p* = 0.005). In addition, the TyG index tended to correlate with ESR (*p* = 0.104), but it was not correlated with BVAS, FFS, or CRP (Appendix A).

### 3.3. Comparison of Cumulative Survival Rates According to the TyG Index

We conducted the ROC curve analysis to obtain the cut-off value of the TyG index for each poor prognosis including all-cause mortality, relapse, ESKD, CVA, and ACS; however, we could not obtain optimal cut-offs (Appendix A). For these reasons, we classified AAV patients into two groups based on 9.011, which was the lower limit of the highest tertile of the TyG index for ACS [12]. When AAV patients were divided into two groups according to a TyG index of 9.011, AAV patients with a TyG index ≥ 9.011 exhibited a significantly lower cumulative ACS-free survival rate than those with a TyG index < 9.011 among five poor prognoses (*p* = 0.046). However, there were no significant differences in the cumulative survival rates of other poor prognoses between the two groups (Figure 1).

### 3.4. Cox Hazards Model Analyses for ACS Occurrence

In the univariable Cox analysis, male sex, BVAS, FFS, T2DM, hypertension, blood urea nitrogen, and a TyG index ≥ 9.011 at AAV diagnosis were associated with the occurrence of ACS during follow-up (*p* < 0.1). In the multivariable Cox analysis, only male sex was found to be a significant independent factor for anticipating the occurrence of ACS in AAV patients (HR 5.548, 95% CI 1.254, 24.541). However, a TyG index ≥ 9.011 could not predict ACS independently (Table 2).

### 3.5. TyG Index Estimating MetS

When the cut-off of the TyG index was set as 8.688 using the ROC curve for the presence of MetS, the sensitivity and specificity for diagnosing MetS were 74.4% and 64.3% (area under the curve [AUC] 0.727, 95% confidence interval [CI], 0.646, 0.808), respectively. When AAV patients were divided into two groups according to a TyG index of 8.688, MetS was identified more frequently in AAV patients with a TyG index ≥ 8.688 than in those with a TyG index < 8.688 (70.9% vs. 31.8%, *p* < 0.001). Furthermore, AAV patients with a TyG index ≥8.688 exhibited a significantly higher risk for MetS than those with a TyG index < 8.688 (RR 5.299, 95% CI 2.606, 10.491) (Figure 2A,B). We also investigated whether a TyG index ≥ 9.011, which was a significant cut-off for the frequency of ACS during follow-up, could estimate MetS. When AAV patients were divided into two groups according to a TyG index of 9.011, MetS was identified more frequently in AAV patients with a TyG index ≥ 9.011 than in those with a TyG index < 9.011 (70.8% vs. 46.2%, *p* = 0.005). Furthermore, AAV patients with a TyG index ≥ 9.011 exhibited a significantly higher risk for MetS than those with a TyG index < 9.011 (RR 2.833, 95% CI 1.363, 5.892) (Figure 2C).

### 3.6. Logistic Regression Analyses for MetS

In the univariable logistic regression analysis, age, BMI, MPO-ANCA (or P-ANCA) positivity, BVAS, FFS, white blood cell count, haemoglobin, platelet count, serum albumin, ESR, CRP, and the TyG index were associated with MetS at AAV diagnosis (*p* < 0.1). Moreover, in the multivariable analysis, BMI (OR 1.327, 95% CI 1.131, 1.556), haemoglobin (OR 0.723, 95% CI 0.538, 0.971), and the TyG index (OR 5.667, 95% CI 2.071, 15.506) were significantly associated with MetS at AAV diagnosis (Appendix A).

### 3.7. BVAS Anticipating ACS Occurrence

We also investigated whether AAV activity based on BVAS could predict ACS occurrence. The optimal cut-off of BVAS for ACS occurrence was obtained as 11.5 using the ROC curve (sensitivity 91.7% and specificity 48.2%, area 0.724, 95% CI 0.579, 0.869). The occurrence of ACS was more frequent in AAV patients with BVAS ≥ 11.5 than in those with BVAS < 11.5 (13.4% vs. 1.5%, *p* = 0.008). Furthermore, AAV patients with BVAS ≥ 11.5 exhibited a significantly higher risk for the occurrence of ACS than those with BVAS < 11.5 (RR 10.225, 95% CI 1.258, 81.389) (Figure 2D,E).

### 3.8. Comparison of Cumulative Survival Rates According to MetS and BVAS

When AAV patients were divided into two groups according to the presence of MetS, AAV patients with MetS exhibited a significantly lower cumulative ACS-free survival rate than those without MetS (*p* = 0.007). AAV patients were also divided into two groups based on BVAS ≥11.5, AAV patients with BVAS ≥ 11.5 exhibited a significantly lower cumulative ACS-free survival rate than those with BVAS < 11.5 (*p* = 0.005) (Figure 3). In addition, we divided AAV patients into four groups according to the presence of MetS and BVAS ≥ 11.5 as follows: group 1 was defined as AAV patients with MetS (+) and BVAS ≥ 11.5; group 2 as those with MetS (+) and BVAS < 11.5; group 3 as those with MetS (−) and BVAS ≥ 11.5; group 4 as those with MetS (−) and BVAS < 11.5. When the incidence rates of ACS were compared among the four groups, AAV patients in group 1 exhibited a significantly lower cumulative ACS-free survival rate than those in the remaining three groups (Appendix A).

## 4. Discussion

In this study, we investigated whether the TyG index at AAV diagnosis could predict the occurrence of ACS during follow-up in AAV patients and obtained several interesting findings. First, AAV patients with the highest tertile of the TyG index at AAV diagnosis exhibited a significantly lower cumulative ACS-free survival rate than the others. However, the highest tertile of the TyG index could not predict ACS independently, and its statistical significance did not exceed that of the male sex in the Cox analysis. Second, the TyG index independently estimated concurrent MetS in AAV patients. Third, AAV patients with BVAS ≥ 11.5 showed a significantly higher RR for ACS compared to patients in the lower BVAS group. Lastly, AAV patients with concurrent MetS and BVAS ≥ 11.5 exhibited significantly lower cumulative ACS-free survival rates than those without. Therefore, we conclude that various factors such as IR and MetS reflected by the TyG index, BVAS, and other conventional risks are involved in the occurrence of ACS.

In this study, we demonstrated that the TyG index could be used to estimate concurrent MetS at AAV diagnosis. This has two meanings. First, we verified that our study population showed clinical characteristics similar to the population to which the TyG index had been applied in previous studies, and that the clinical implication of TyG in AAV patients may be similar to that in the general population [21]. The other is that the TyG index may ultimately estimate an increase in the risk of coronary artery disease in AAV patients through MetS as one of the mechanistic hypotheses that can lead to ACS [22]. To prove this assumption, we investigated the relationship between the TyG index and the presence of MetS frequency at AAV diagnosis. We applied a TyG index ≥ 9.011, which was a significant cut-off for the occurrence of ACS during follow-up. AAV patients with a TyG index ≥ 9.011 had MetS more frequently and exhibited a significantly lower cumulative ACS-free survival rate than those with a TyG index < 9.011. Therefore, although the exact mechanism remains uncertain, we conclude that the TyG index at diagnosis can predict the occurrence of ACS during follow-up in AAV patients by easily assessing concomitant metabolic abnormalities that may contribute to the occurrence of coronary arterial disease in general.

ACS, which occurs in the general population with IR represented by elevated levels of fasting plasma glucose and TG, is mainly characterised by chronic atherosclerotic lesions [13,23,24]. Therefore, more aggressive correction and closer follow-up of IR in the general population have a benefit of lowering ACS frequency. Conversely, ACS in AAV patients is more frequently associated with acute thrombotic events due to vasculitis itself and increased vasculitis activity in addition to chronic atherosclerosis lesions compared to the general population. Therefore, with improvement of IR, control of AAV activity is needed to prevent the occurrence of ACS during the follow-up. A previous article reported an interesting case of acute MI without evidence of typical advanced atherosclerotic lesions, including necrotic core, in a patient with MPO-ANCA-associated vasculitis [25]. This case may support our assumption of the association of ACS with not only IR and MetS but also inflammation related to vasculitis itself. To prove this assumption, we investigated the effect of BVAS on the occurrence of ACS. When the multivariable Cox analysis included BVAS as a continuous variable, BVAS could not predict ACS significantly (Table 2). Also, when BVAS ≥ 11.5 was included in the multivariable Cox analysis instead of BVAS, BVAS ≥ 11.5 could not predict ACS independently in the multivariable Cox analysis (Appendix A). Nevertheless, we found that the occurrence of ACS was more frequent in AAV patients with BVAS ≥ 11.5 than those with BVAS < 11.5. Furthermore, AAV patients with BVAS ≥ 11.5 exhibited a significantly lower cumulative ACS-free survival rate than those with BVAS < 11.5. Therefore, we conclude that AAV activity, in addition to IR and MetS, may contribute to the occurrence of ACS during follow-up in AAV patients.

MetS has been considered as an important risk factor for cardiovascular disease even in patients with ANCA-associated vasculitis. Our research group had also demonstrated that MetS was associated with poor health outcomes including ACS in patients with MPO-ANCA-associated vasculitis [26]. However, measuring waist circumference is an operator-dependent examination. Thus, an easier and more precise tool such as TyG index in addition to MetS is desired in actual clinical settings. Furthermore, due to the dichotomous nature of MetS, it cannot be quantified and is hard to track over time to assess the changes in insulin resistance status and overall metabolic risk. Moreover, insulin resistance is a main pathogenic factor for MetS and cardiovascular disease [27]. As insulin resistance usually occurs prior to MetS, the assessment of insulin resistance status in individuals may be more important in clinical practice. However, assessing conventional insulin resistance requires a special laboratory which is not available in most clinical settings. The TyG index has been suggested as a reliable surrogate marker of insulin resistance, and gives us insight on prediction of cardiovascular disease in the general population [12,28]. In the present study, we demonstrated the predictive ability of TyG for MetS and the close association between TyG, MetS and cardiovascular disease in patients with AAV. Therefore, considering the fact that serum glucose and TG level is easy to obtain and the calculation of TyG index is simple in clinic, it is an applicable index for most physicians as opposed to MetS.

The overall hypotheses are summarised in Figure 4. From other previous studies, IR and MetS are already known as a risk factor for ACS occurrence [10]. Also, TyG index and MetS is known as a surrogate marker for IR and reflects the severity of metabolic abnormalities in individuals [12,13]. Next, IR and MetS subsequently provoke chronic atherosclerotic lesions in the coronary arteries and ultimately contribute to the occurrence of ACS. Therefore, the TyG index may predict the occurrence of ACS in AAV patients. Second, in terms of the occurrence of ACS demonstrated by AAV activity based on BVAS, high BVAS induces not only chronic atherosclerotic lesions but also acute thrombotic lesions in coronary arteries, in proportion to the inflammatory burden. There was no direct correlation between the TyG index and BVAS at diagnosis, whereas the TyG index tended to correlate with ESR. Therefore, we assumed that the TyG index may predict the occurrence of ACS during follow-up by reflecting the extent of inflammation in AAV patients. Third, chronic inflammation can increase the severity of IR, which may accelerate and aggravate the progression to ACS in AAV patients [29].

This is the first study to elucidate the clinical implications of the TyG index in estimating concurrent MetS and its anticipation of the occurrence of ACS in AAV patients. Furthermore, we hypothesised that the TyG index could predict future ACS in AAV patients.

### Limitations

The small number of AAV patients did not allow for subgroup analyses according to AAV subtypes. In particular, it was impossible to determine the association between the TyG index and the patterns of coronary arterial lesions in 12 AAV patients with ACS. The retrospective study design did not permit direct measurement of IR, and furthermore, it resulted in the omission of patients whose medical records were not sufficient to know whether metabolic syndrome could be classified; hence, we could not reveal the effect of IR on ACS in AAV patients. Another limitation is that there were no data regarding specific treatments that may directly influence overall cholesterol levels and cardiovascular disease risks such as anti-dyslipidemia agents or anti-platelet agents. Moreover, we also could not collect data on smoking history among study participants. A prospective future study with a larger number of AAV patients will compensate for these limitations and provide more reliable information on the application to AAV patients in real clinical settings.

## 5. Conclusions

The TyG index at AAV diagnosis could estimate the concurrent presence of MetS and predict the occurrence of ACS during follow-up along with high BVAS at diagnosis in patients with AAV. Therefore, we suggest that the TyG index at the time of AAV diagnosis may be calculated for all patients, if possible. Furthermore, when a patient has a TyG index within its highest tertile at AAV diagnosis, close attention should be paid to these patients, and frequent visits should be recommended.

## Figures and Tables

**Figure 1 diagnostics-12-01486-f001:**
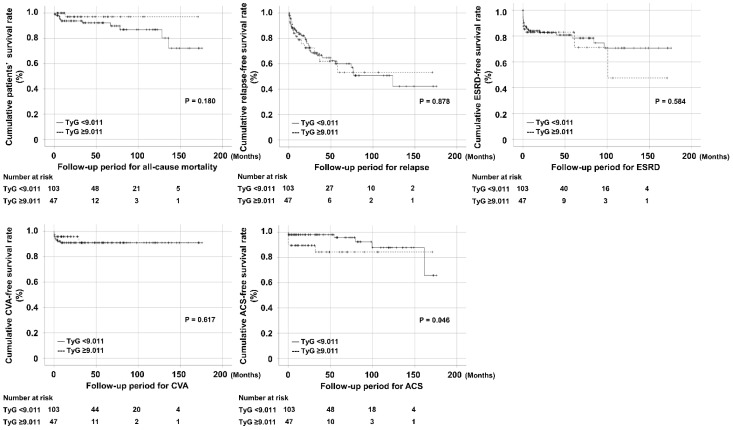
Comparison of cumulative survival rates according to the TyG index. Patients were divided into two groups according to the highest tertile of the TyG index (≥9.011). Only the occurrence of ACS significantly differed between AAV patients with a TyG index ≥ 9.011 and those with a TyG index < 9.011 among the five poor prognoses. TyG, triglyceride-glucose; ACS, acute coronary syndrome; AAV, ANCA-associated vasculitis; ANCA, antineutrophil cytoplasmic antibody; ESKD, end-stage renal disease; CVA, cerebrovascular accident.

**Figure 2 diagnostics-12-01486-f002:**
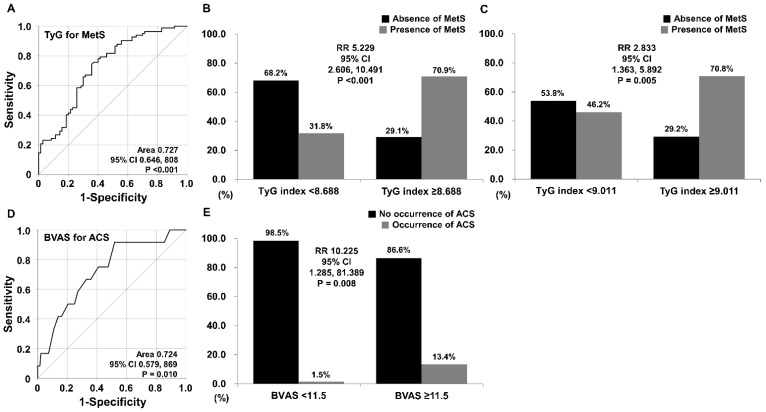
TyG index estimating MetS and BVAS anticipating ACS. (**A**) The cut-off value of the TyG index for the presence of MetS was obtained using the ROC curve and was set as 8.688. (**B**) AAV patients with a TyG index ≥ 8.688 more often had MetS than those with a TyG index < 8.688. (**C**) When the cut-off TyG index of 9.011 was applied, AAV patients with a TyG index ≥ 9.011 more frequently had MetS than those with a TyG index < 9.011. (**D**) The cut-off of BVAS for ACS occurrence was obtained using the ROC curve and was set as 11.5. (**E**) AAV patients with BVAS ≥ 11.5, exhibited a significantly higher risk for the occurrence of ACS than those with BVAS < 11.5. TyG, triglyceride-glucose; MetS, metabolic syndrome; ROC, receiver operating characteristic; AAV, ANCA-associated vasculitis; ANCA, antineutrophil cytoplasmic antibody; BVAS, Birmingham vasculitis activity score; ACS, acute coronary syndrome.

**Figure 3 diagnostics-12-01486-f003:**
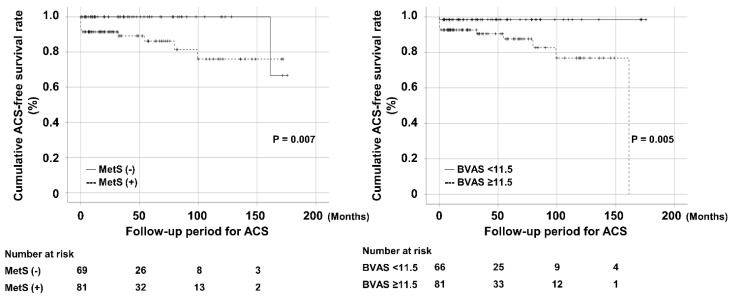
Comparison of cumulative survival rates according to MetS and BVAS. AAV patients with MetS exhibited a significantly lower cumulative ACS-free survival rate than those without MetS, and AAV patients with BVAS ≥ 11.5, exhibited a significantly lower cumulative ACS-free survival rate than those with BVAS < 11.5. MetS, metabolic syndrome; BVAS, Birmingham vasculitis activity score; AAV, ANCA-associated vasculitis; ANCA, antineutrophil cytoplasmic antibody; ACS, acute coronary syndrome.

**Figure 4 diagnostics-12-01486-f004:**
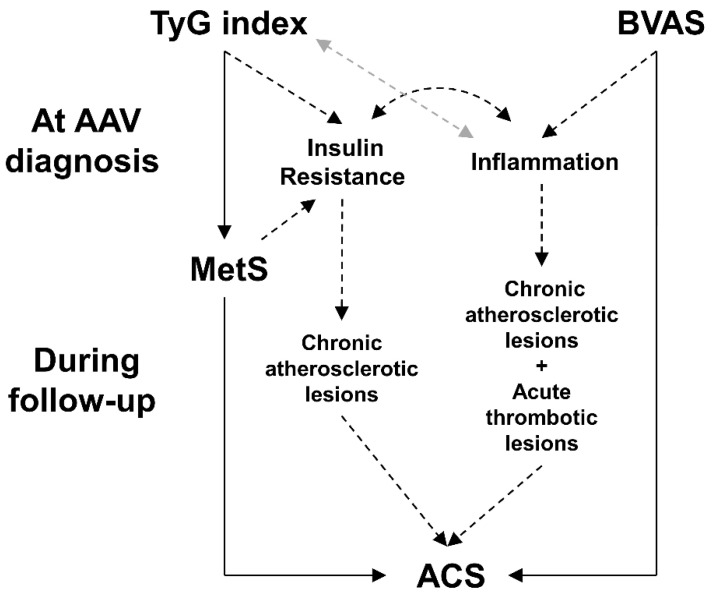
Overall hypotheses of a mechanism from the TyG index to ACS. Black solid arrow: the results of this study; black dotted arrow: hypothesis with high probability; grey dotted arrow: hypothesis with low probability. AAV, ANCA-associated vasculitis; ANCA, antineutrophil cytoplasmic antibody; TyG, triglyceride-glucose; MetS, metabolic syndrome; BVAS, Birmingham vasculitis activity score; ACS, acute coronary syndrome.

**Table 1 diagnostics-12-01486-t001:** Characteristics of AAV patients (N = 152).

Variables	Values
* **At diagnosis** *	
**Basic clinical data**	
Age (years)	57.2 (20.9)
Male sex (N, (%))	50 (32.9)
BMI (kg/m^2^)	22.1 (4.6)
**AAV subtypes (N, (%))**	
MPA	84 (55.3)
GPA	35 (23.0)
EGPA	33 (21.7)
**ANCA positivity (N, (%))**	
MPO-ANCA (or P-ANCA) positive	102 (67.1)
PR3-ANCA (or C-ANCA) positive	22 (14.5)
Both ANCA positive	5 (3.3)
ANCA positive	119 (78.3)
**AAV-specific indices**	
BVAS	12.0 (12.0)
FFS	1.0 (2.0)
**Comorbidities (N, (%))**	
T2DM	44 (28.9)
Hypertension	46 (30.3)
**Laboratory results**	
White blood cell count (/mm^3^)	8640.0 (6835.0)
Haemoglobin (g/dL)	11.4 (4.0)
Platelet count (×1000/mm^3^)	286.0 (158.0)
Blood urea nitrogen (mg/dL)	18.9 (23.0)
Serum creatinine (mg/dL)	1.0 (1.4)
Serum albumin (g/dL)	3.6 (0.9)
Total cholesterol (mg/dL)	171.5 (65.0)
HDL-cholesterol (mg/dL)	48.0 (25.0)
LDL-cholesterol (mg/dL)	91.4 (47.0)
**Acute phase reactants**	
ESR (mm/h)	56.0 (66.0)
CRP (mg/L)	7.0 (66.7)
**TyG index-related variables**	
Fasting glucose (mg/dL)	103.0 (36.0)
Triglyceride (mg/dL)	114.5 (74.0)
**TyG index**	8.8 (0.7)
* **During follow-up** *	
**Poor prognoses (N, (%))**	
All-cause mortality	12 (7.9)
Relapse	48 (31.6)
ESKD	31 (20.4)
CVA	12 (7.9)
ACS	12 (7.9)
**Follow-up period for each poor prognosis (months)**	
All-cause mortality	33.4 (66.0)
Relapse	20.1 (42.8)
ESKD	20.8 (62.1)
CVA	30.0 (60.8)
ACS	32.6 (63.4)
**Medications (N, (%))**	
Glucocorticoids	144 (94.7)
Cyclophosphamide	78 (51.3)
Rituximab	24 (15.8)
Mycophenolate mofetil	23 (15.1)
Azathioprine	76 (50.0)
Tacrolimus	11 (7.2)
Methotrexate	15 (9.9)
Plasma exchange	10 (6.6)

Values are expressed as a median (interquartile range, IQR) or N (%). AAV, ANCA-associated vasculitis; ANCA, antineutrophil cytoplasmic antibody; BMI, body mass index; MPA, microscopic polyangiitis; GPA, granulomatosis with polyangiitis; EGPA, eosinophilic granulomatosis with polyangiitis; MPO, myeloperoxidase; P, perinuclear; PR3, proteinase 3; C, cytoplasmic; BVAS, Birmingham vasculitis activity score; FFS, five-factor score; T2DM, type 2 diabetes mellitus; HDL, high density lipoprotein; LDL, low density lipoprotein; ESR, erythrocyte sedimentation rate; CRP, C-reactive protein; TyG, triglyceride and glucose; ESKD, end-stage renal disease; CVA, cerebrovascular accident; ACS, acute coronary syndrome.

**Table 2 diagnostics-12-01486-t002:** Cox hazards model analysis of variables at AAV diagnosis for ACS during follow-up.

Variables	Univariable			Multivariable		
	HR	95% CI	*p* Value	HR	95% CI	*p* Value
Age	1.024	0.978, 1.071	0.309			
Male sex	4.933	1.403, 17.345	0.013	5.548	1.254, 24.541	0.024
BMI	1.131	0.940, 1.359	0.192			
MPA	1.595	0.475, 5.354	0.450			
GPA	0.730	0.159, 3.340	0.685			
EGPA	0.674	0.144, 3.151	0.616			
MPO-ANCA (or P-ANCA) positive	2.930	0.622, 13.804	0.174			
PR3-ANCA (or C-ANCA) positive	1.954	0.249, 15.314	0.524			
BVAS	1.120	1.033, 1.215	0.006	1.099	0.972, 1.242	0.131
FFS	1.936	1.147, 3.265	0.013	1.357	0.722, 2.550	0.343
T2DM	4.255	1.245, 14.537	0.021	1.583	0.411, 6.093	0.504
Hypertension	3.086	0.971, 9.805	0.056	2.748	0.796, 9.485	0.110
White blood cell count	1.000	1.000, 1.000	0.320			
Haemoglobin	0.807	0.615, 1.059	0.123			
Platelet count	1.000	0.997, 1.004	0.842			
Blood urea nitrogen	1.013	1.000, 1.027	0.054	0.997	0.978, 1.017	0.800
Serum creatinine	1.142	0.911, 1.432	0.249			
Serum albumin	0.600	0.273, 1.320	0.204			
Total cholesterol	1.005	0.992, 1.017	0.456			
HDL-cholesterol	0.986	0.954, 1.019	0.401			
LDL-cholesterol	1.007	0.993, 1.020	0.318			
ESR	1.007	0.992, 1.022	0.350			
CRP	1.005	0.997, 1.013	0.249			
TyG ≥ 9.011	3.054	0.959, 9.726	0.059	2.312	0.576, 9.281	0.237

AAV, ANCA-associated vasculitis; ANCA, antineutrophil cytoplasmic antibody; ACS, acute coronary syndrome; BMI, body mass index; MPA, microscopic polyangiitis; GPA, granulomatosis with polyangiitis; EGPA, eosinophilic granulomatosis with polyangiitis; MPO, myeloperoxidase; P, perinuclear; PR3, proteinase 3; C, cytoplasmic; BVAS, Birmingham vasculitis activity score; FFS, five-factor score; HDL, high density lipoprotein; LDL, low density lipoprotein; ESR, erythrocyte sedimentation rate; CRP, C-reactive protein; TyG, triglyceride and glucose.

## Data Availability

The raw data supporting the conclusions of this article will be made available by the authors, without undue reservation.

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
