# Peer review of "Triglyceride and Glucose Index Predicts Acute Coronary Syndrome in Patients with Antineutrophil Cytoplasmic Antibody-Associated Vasculitis"

_diagnostics, 2022, doi:10.3390/diagnostics12061486_

Round 1

Reviewer 1 Report

The authors investigated the predictive factor of cardiovascular events in AAV patients. In particular, the authors focused on TyG index. Although the manuscript is well-written, the finding of main point that the authors want to clarify is weak. So, I would like to describe several comments to improve article. 

What is the importance of TyG index? I think the evaluation of Mets is more important than TyG index after reading current manuscript. So, I think it is difficult to recommend publication this manuscript if the authors would describe the importance of TyG index. 

I think the authors have found more important issue in this manuscript. According to the manuscript, mechanisms of developing cardiovascular events in AAV patients are both disease activity of AAV (BVAS) and patients’ atherosclerotic conditions (Mets).

So, for example, I recommend that the authors divide this study population into four groups by with or without Mets and high or low BVAS, respectively. And then compare the incidence of cardiovascular disease among four groups.

If the authors could find good results, the authors may re-write the manuscript according to the results.
Anyway, authors should change the focusing point in the manuscript because clinical significance of TyG index is very weak. 

Reviewer 2 Report

Pil Gyu Park et al performed a retrospective analysis on the triglyceride and glucose index as a predictor of acute coronary syndrome in patients with ANCA vasculitis (AAV). 

The TyG index is closely associated with traditional cardiovascular risk factors in general populations.

I think that the topic of the article is interesting because this TyG index could be useful for identification of patients with AAV with higher risk of cardiovascular disease.

The manuscript is an Original article and respects the format requirements of this type of article. I think that it is concise, clearly and well written. Also, the size of the patient cohort studied is substantially large. 

In the abstract authors summarize the study. 

Introduction, Methods, Results and Discussion are clearly exposed.

I have only one minor observation: according to the new nomenclature of kidney function and kidney diseases (Levey 2020, DOI: 10.1093/eurheartj/ehaa650), the authors should modify the term renal with kidney (ESKD)

Round 2

Reviewer 1 Report

I understood that the authors do want to focus on TyG index.

However, I think it is not so difficult for clinical physicians to diagnose Mets. 

In addition, although TyG index did not significantly associate with clinical outcomes, Mets significantly associated with development of CVDs in this study. 

So, Mets but not TyG index is the useful factor of predicting CVD in AAV patients, and because diagnosing Mets is not so hard, significance of TyG should be low. 

Therefore, authors should demonstrate the significance of TyG index in clinical setting. 

Author Response

Reviewer’s comments

Manuscript number: diagnostics-1695880

Title: Triglyceride and glucose index predicts acute coronary syndrome in patients with antineutrophil cytoplasmic antibody-associated vasculitis

1) I understood that the authors do want to focus on TyG index. However, I think it is not so difficult for clinical physicians to diagnose Mets.

We understand that diagnosing metabolic syndrome is not that difficult in most clinical settings around the world. We were focused on clinical situations in South Korea where time is very limited such that measuring waist circumference of patients is not practical in many cases. We agree that diagnosis of metabolic syndrome is applicable in general. However, we think that measuring waist circumference can be inconsistent depending on physicians even with controlled protocols. We believe that TyG index can provide an easy and precise data to present meaningful insights on AAV prognosis in addition to metabolic syndrome. We made changes to main manuscript as written below.

“MetS has been considered as an important risk factor for cardiovascular disease even in patients with ANCA-associated vasculitis. Our research group also had demonstrated that MetS was associated with poor health outcome including ACS in patients with MPO-ANCA-associated vasculitis [26]. However, measuring waist circumference is an operator-dependent examination. Thus an easier and more precise tool such as TyG index in addition to MetS is desired in actual clinical settings. Furthermore, due to the dichotomous nature of MetS, it cannot be quantified and is hard to track over time to assess the changes in insulin resistance status and overall metabolic risk. Moreover, insulin resistance is a main pathogenic factor for MetS and cardiovascular disease [27]. As insulin resistance usually occurs prior to MetS, the assessment of insulin resistance status in individuals may be more important in clinical practice. However, assessing conventional insulin resistance requires special laboratory which is not available in most clinical settings. TyG index has been suggested as a reliable surrogate marker of insulin resistance, and gives us insight on prediction of cardiovascular disease in the general population [28, 29]. In the present study, we demonstrated predictive ability of TyG for MetS and the close association between TyG, MetS and cardiovascular disease in patients with AAV. Therefore, considering the fact that serum glucose and TG level is easy to obtain and the calculation of TyG index is simple in clinic, it is applicable index to most physicians rather than MetS.” (Lines 334-352)

2) In addition, although TyG index did not significantly associate with clinical outcomes, Mets significantly associated with development of CVDs in this study. So, Mets but not TyG index is the useful factor of predicting CVD in AAV patients, and because diagnosing Mets is not so hard, significance of TyG should be low. Therefore, authors should demonstrate the significance of TyG index in clinical setting. 

Thank you for the comments, and we sincerely feel that the comments you gave are meaningful to our study. As you pointed out, we have provided an evidence of metabolic syndrome as a good predictor for CVD in AAV patients. In fact, we have strengthened the point which was already revealed in a study published early this year (Lee et al. Rheumatol Int. 2022, doi: 10.1007/s00296-021-04908-1). Therefore, we believe that instead of focusing on metabolic syndrome, which studies are previously proceeded, investigating the role of TyG index is clinically important.

The association between metabolic syndrome in AAV and poor outcomes including CVD has been elucidated previously in several studies. Citations are listed listed below.

  1. Lee, S.B.; Kwon, H.C.; Kang, M.I.; Park, Y.B.; Park, J.Y.; Lee, S.W.; Increased prevalence rate of metabolic syndrome is an independent predictor of cardiovascular disease in patients with antineutrophil cytoplasmic antibody-associated vasculitis. Rheumatol Int. 2022, 42, 291-302. doi: 10.1007/s00296-021-04908-1.
  2. Park, P.G.; Pyo, J.Y.; Ahn, S.S.; Song, J.J.; Park, Y.B.; Huh, J.H.; Lee, S.W.; Metabolic Syndrome Severity Score, Comparable to Serum Creatinine, Could Predict the Occurrence of End-Stage Kidney Disease in Patients with Antineutrophil Cytoplasmic Antibody-Associated Vasculitis. J Clin Med. 2021, 8, 5744. doi: 10.3390/jcm10245744.
  3. Park, P.G.; Pyo, J.Y.; Ahn, S.S.; Song, J.J.; Park, Y.B.; Huh, J.H.; Lee, S.W.; Effect of numbers of metabolic syndrome components on mortality in patients with antineutrophil cytoplasmic antibody-associated vasculitis with metabolic syndrome. Clin Exp Rheumatol. 2022, 40, 758-764, doi: 10.55563/clinexprheumatol/k4m3it.

Due to the nature of single center study and limited number of enrolled patients, we are aware of the fact that statistical strength is not enough. We have listed this limitation in DISCUSSION section. We believe that if the study population is large enough with multicenter enrollment, the optimal cutoff of TyG for predicting ACS will be meaningful in multivariable Cox analysis. Our study is significant in a way that it is a first pilot study that TyG, a marker for insulin resistance and NAFLD, can be used to measure potential risk for ACS in AAV patients.

Thank you for your consideration.

Round 3

Reviewer 1 Report

Although I understood that the authors want to emphasize the significance of TyG, only way is to demonstrate the association between TyG and clinical outcomes or atherosclerotic surrogate markers.  Otherwise, clinical importance of TyG is still low.